# Chemical Composition and Antistaphylococcal Activity of Essential Oil of *Curcuma mangga* Rhizome from Indonesia

Klara Urbanova [1], Andreas Romulo [2], Marketa Houdkova [3], Pavel Novy [4] and Ladislav Kokoska [3,*]

[1] Department of Sustainable Technologies, Faculty of Tropical AgriSciences, Czech University of Life Sciences Prague, Kamycka 129, 165 00 Prague, Czech Republic; urbanovak@ftz.czu.cz
[2] Food Technology Department, Faculty of Engineering, Bina Nusantara University, Jakarta 11480, Indonesia; andreas.romulo@binus.edu
[3] Department of Crop Sciences and Agroforestry, Faculty of Tropical AgriSciences, Czech University of Life Sciences Prague, Kamycka 129, 165 00 Prague, Czech Republic; houdkovam@ftz.czu.cz
[4] Department of Food Science, Faculty of Agrobiology, Food and Natural Resources, Czech University of Life Sciences Prague, Kamycka 129, 165 00 Prague, Czech Republic; novy@af.czu.cz
* Correspondence: kokoska@ftz.czu.cz

**Abstract:** This study assessed the antistaphylococcal activity of essential oil (EO) hydrodistilled from the rhizome of *Curcuma mangga* grown in Indonesia using the broth microdilution volatilization method and standard broth microdilution method modified for evaluation of volatile agents, as well as described its chemical composition using gas chromatography (GC) with mass spectrometry (MS). A fused-silica HP-5MS column and a DB-17MS column were used to separate the components into two columns. The results demonstrated that the EO exhibited antistaphylococcal activity at the minimum inhibitory concentration (MIC) ranging from 128 to 1024 μg/mL. In contrast, the clinical isolate of tetracycline-resistant *Staphylococcus aureus* was the most sensitive strain (MIC 128 μg/mL). The major constituents of the EO were 15,16-dinorlabda-8(17),11-dien-13-one (24.63/15.78%), followed by ambrial (16.12/10.97%), 13-nor-eremophil-1(10)-en-11-one (7.16/6.21%), 15,16-dinorlabda-8(17),12-dien-14-al (6.61/11.57%), and aromadendrene oxide (5.98/3.77%). These results propose *C. mangga* rhizome EO as a promising agent for developing natural-based anti-infective preparations.

**Keywords:** antibacterial activity; *Curcuma mangga*; essential oil; GC-MS; *Staphylococcus aureus*

## 1. Introduction

Infectious diseases continue to pose significant challenges to public health, particularly in low- and middle-income countries, where the escalating prevalence of microorganisms resistant to antimicrobial agents is a pressing concern [1,2]. Among these pathogens, *Staphylococcus aureus* stands out as a human threat, contributing substantially to the burden of morbidity associated with community and hospital-acquired infections. These infections range from lower respiratory tract infections to pneumonia and nosocomial bacteremia [3,4].

Historically, *S. aureus* infections were effectively treated with β-lactam antibiotics [5]. However, the emergence of resistant strains, particularly methicillin-resistant *S. aureus* (MRSA), has become a significant impediment to successful treatment [6]. The evolution of resistance mechanisms, including the expression of enzymes that hydrolyze the β-lactam ring and the acquisition of the mecA gene, encoding a modified penicillin-binding protein 2a, has rendered conventional antibiotics ineffective [7]. The urgent need for novel therapeutic agents capable of combating staphylococcal infections has become paramount in the face of this escalating challenge.

Researchers have increasingly focused on plant-derived products, such as extracts and essential oils (EOs), as potential alternatives for treating staphylococcal infections. While these natural remedies have a long history of use in traditional medicine for infectious diseases, the renewed interest in EOs gained momentum in the late 20th century [8,9].

Notably, the essential oil derived from the Australian native tree *Melaleuca alternifolia* (Myrtaceae) has demonstrated significant antimicrobial effects against *S. aureus* and MRSA in both in vitro studies and clinical trials [10–12].

Plants serve as vast repositories of biochemical compounds, particularly secondary metabolites, which are produced as a defence mechanism against various stressors, including pathogens. These secondary metabolites have played a pivotal role in traditional medicine, forming the basis of remedies for infectious diseases across different cultures and eras. The historical significance of medicinal plants is evident in ancient texts from Egypt, China, and the Mediterranean, reflecting their enduring role in human well-being.

In the contemporary era, traditional herbal preparations remain a primary source of medicine, with approximately 80% of people in developing countries relying on these remedies. The World Health Organization (WHO) acknowledges the significant contribution of traditional medicine to global healthcare [13,14].

Despite the remarkable progress in chemical synthesis, around 25% of drugs currently used in modern medicine can be traced back to plant origins, particularly in the categories of anticancer and anti-infective drugs [15]. The drug discovery process often involves the exploration of plant species, and three main approaches guide the selection: random, chemotaxonomical, and ethnopharmacological [16]. The random approach entails the collection of plants without specific criteria, aiming to capture a diverse range of chemical compounds. The chemotaxonomical selection focuses on the presence of markers within plant families or genera known for specific biological activities. Ethnopharmacological selection relies on traditional medicinal knowledge, with oral or written information guiding the identification of plants for further investigation [17].

Ethnopharmacology, a multidisciplinary approach, integrates observations, experimental investigations, and traditional knowledge to identify plants with bioactive compounds. This approach involves gathering information from various sources, including traditional medical systems and diverse cultural practices. The selected plants are then subjected to extraction and antimicrobial testing, demonstrating the potential of these natural products in developing anti-infective agents. The exploration of plant-derived compounds continues to be a cornerstone in drug discovery, with traditional medicine providing valuable insights. The diverse approaches in plant selection, coupled with advancements in scientific methodologies, contribute to the ongoing search for novel anti-infective agents. This research aligns with the global efforts to address the challenges posed by infectious diseases and antibiotic-resistant pathogens.

Indonesia, recognized as a leading producer of economically significant essential oils, boasts a rich diversity of plant species with therapeutic potential. EOs derived from clove (*Syzygium aromaticum*), nutmeg (*Myristica fragrans*), and patchouli (*Pogostemon cablin*) have long been exploited for their medicinal properties [18]. The family Zingiberaceae, represented by 19 genera and 375 species, is particularly noteworthy. This plant family is frequently employed in traditional Indonesian medicine known as Jamu [19]. Numerous studies have highlighted the bioactive nature of EOs within the Zingiberaceae family, showcasing their significant antimicrobial effects [20–22]. *Curcuma* spp., one of the largest genera within the Zingiberaceae family, comprises approximately 80 species distributed widely in Indonesia. These plants have played a multifaceted role as foods, spices, and medicines in traditional practices, offering potential therapeutic benefits against various diseases [23,24].

Among these, *Curcuma mangga* Valeton & Zijp, commonly known as mango ginger, stands out. This erect, tillering, herbaceous perennial is characterized by an underground rhizome that emanates a distinctive mango fragrance when sliced or crushed. Widely distributed from Indonesia (Java) to the Malay Peninsula and Thailand, *C. mangga* is frequently employed in traditional medicine to alleviate stomach complaints, gastric ulcers, chest pain, fever, and general debility [25,26].

Previous studies on *C. mangga* rhizome EO, particularly from Malaysia, have reported predominantly monoterpenes such as myrcene and β-pinene [27]. In contrast, Kamaz-

eri [28] identified sesquiterpenes, including caryophyllene and caryophyllene oxide, as major compounds, demonstrating their antimicrobial efficacy against *S. aureus* and *C. albicans*. Despite these findings, there remains a notable gap in our understanding, particularly regarding the EO derived from *C. mangga* rhizome grown in Indonesia. Limited reports exist on its chemical composition and antimicrobial activity against a broad spectrum of standard and clinical strains of *S. aureus*.

This research endeavours to bridge this knowledge gap by comprehensively investigating the chemical composition and antistaphylococcal effects of *C. mangga* rhizome EO from Indonesia. Through detailed analyses and experiments, we aim to uncover the potential of this essential oil as a valuable resource in the quest for novel therapeutic agents against staphylococcal infections. Therefore, this research determined the chemical composition and antistaphylococcal effect of *C. mangga* rhizome EO from Indonesia and the implications of this research, showing the promising role of *C. mangga* rhizome EO in antimicrobial therapeutics.

## 2. Materials and Methods

### 2.1. Plant Materials and Sample Preparation

*C. mangga* (Voucher specimen no. Ar-0069) was obtained from the Biopharmaca collection garden at Bogor Agricultural University (IPB) in Dramaga-Bogor (West Java Province, Indonesia). Plant material was dried and homogenized using a Grindomix apparatus (GM 100 Retsch, Haan, Germany).

### 2.2. Hydrodistillation of EO

The essential oil was extracted by hydrodistillation of 50 g of *C. mangga* rhizome powdered samples in 1 L of distilled water for 3 h using a Clevenger apparatus (Merci, Brno, Czech Republic). The collected EO was dried using anhydrous sodium sulphate (Merck, Darmstadt, Germany) and stored at 4 °C in airtight glass vials.

### 2.3. Microorganisms and Media

In the context of this investigation, standard strains from the American Type Culture Collection (ATCC) were employed, including *Candida albicans* ATCC 10231, *Enterococcus faecalis* ATCC 29212, *Escherichia coli* ATCC25922, *Pseudomonas aeruginosa* ATCC 27853, *S. aureus* ATCC 29213, *S. aureus* ATCC 33591, *S. aureus* ATCC 33592, and *S. aureus* ATCC BAA 976. Additionally, clinical isolates comprising methicillin-resistant *S. aureus* (MRSA 1, MRSA 2, MRSA 3, and MRSA 4) and tetracycline-resistant *S. aureus* (TRSA 1 and TRSA 2) were obtained from agar plates sourced from Motol University Hospital in Prague, Czech Republic. Matrix-assisted laser desorption/ionization time-of-flight mass spectrometry, as detailed in Rondevaldova [29], was employed for the identification of clinical isolates. Cultures of microorganisms were maintained on Mueller–Hinton agar (MHA) and stored at 4 °C until use. Before testing, stock cultures were cultivated in Mueller–Hinton broth (MHB) at 37 °C for 24 h, with pH adjusted to 7.6 using Trizma base (Sigma-Aldrich, Prague, Czech Republic). The ATCC strains and media were procured from Oxoid (Basingstoke, UK). To standardize the inoculum, the turbidity of the microorganism suspension was adjusted to the 0.5 McFarland standard ($1.5 \times 10^8$ CFU/mL) using a Densi-La-Meter II (Lachema, Brno, Czech Republic) spectrophotometric device. Tetracycline and tioconazole (Sigma-Aldrich, Prague, Czech Republic) were employed as positive controls.

### 2.4. Antimicrobial Assay

The antibacterial potential of the EO in liquid and vapour phases was subjected to the broth microdilution volatilization method [30]. The antistaphylococcal activity was determined using the standard broth microdilution method according to the Clinical and Laboratory Standards [31] modified by Rondevaldova [32]. The experiments were conducted in white 96-well immunoplates (total well volume = 400 μL) covered by tight-fitting lids with flanges designed to reduce evaporation (SPL Life Sciences, Naechon-Myeon, Re-

public of Korea). Initially, 30 μL of MHA (Oxoid, Basingstoke, UK) was pipetted into every flange on the lid and inoculated with 5 μL of bacterial suspension after agar solidification. Then, the EO was dissolved in 100% DMSO (Sigma-Aldrich, Prague, Czech Republic) and diluted in MHB. The final concentration of DMSO in the microtiter plate wells did not exceed 1%. 100 μL two-fold serial dilution of each sample concentration starting from 1024 μg/mL was distributed into the plates. The plates were then inoculated with bacterial suspensions. Finally, the clamps (Lux Tool, Prague, Czech Republic) with the handmade wooden pads (size 8.5 × 13 × 2 mm) were used for fastening the plate and lid together. An experimental protocol was implemented to determine antistaphylococcal activity; the plates were covered by the EVA Capmat (Micronic, Aston, PA, USA) immediately after inoculating the bacterial suspension to prevent evaporation of the EO. Then, the plates were incubated at 37 °C for 24 h for bacteria (48 h for *C. albicans*). The minimum inhibitory concentration (MIC) values were evaluated by visual assessment of microbial growth after colouring of the metabolically active colonies of microorganisms with thiazolyl blue tetrazolium bromide (MTT) dye (Sigma-Aldrich, Prague, Czech Republic) when the interface of colour changed from yellow to purple (relative to that of the colours in control wells) was recorded. The DMSO at the concentration of 1% did not inhibit any of the strains tested either in the broth or agar media. All experiments were performed in triplicate in three independent experiments, and results were expressed as the median/mode of MICs.

*2.5. Gas Chromatography (GC) and Mass Spectrometry (MS) Analyses*

Gas chromatography (GC) and mass spectrometry (MS) analyses were conducted using an Agilent GC-7890B coupled to a single quadrupole mass selective detector (MSD), i.e., Agilent MSD-5977B (Agilent Technologies, Santa Clara, CA, USA). For the separation of components, two columns were used: a fused-silica HP-5MS column (30 m × 0.25 mm, 0.25 μm, Agilent 19091s-433) and a DB-17MS column (30 m × 0.25 mm, 0.25 μm, Agilent 122-473). The sample was diluted in *n*-hexane (Merck, Darmstadt, Germany) to the concentration of 5 μg/mL, and 1 μL of the solution was injected in split mode (1:20) into the inlet (250 °C), while carrier gas He (1 mL/min) was used. The temperature program of the analyses started at 60 °C (3 min), then shifted to 5 °C/min to 231 °C (10 min). The temperature of the ion source was 200 °C. Standard 70 eV ionization energy spectra were recorded from 30 to 600 *m/z*. Following the same operational parameters, the determination of essential oil compounds was also realized by gas chromatography with a quadrupole time-of-flight mass spectrometer (GC-QTOF-MS, Agilent GC 7890B/QTOF 7200B, Agilent Technologies, Santa Clara, CA, USA) and a fused-silica HP-5MS column (30 m × 0.25 mm, 0.25 μm, Agilent 19091s-433).

The identification of chemical compounds was based on comparing their mass spectra with the National Institute of Standards and Technology Library ver. 2.0.f (NIST, Gaithersburg, MD, USA) [33], retention indices (RI) [34] and co-injection of authentic standard compounds (Sigma-Aldrich, Prague, Czech Republic). The relative percentage contents of essential oil components were determined from FID, indicated on both columns.

**3. Results and Discussion**

The hydrodistilled EO underwent an initial assessment of its antimicrobial properties utilizing the broth microdilution volatilization method [30]. However, with the exception of the oil's growth-inhibitory impact on *S. aureus* in the liquid phase, no discernible positive results were observed for the other tested microorganisms. In response to this, a more detailed evaluation of the EO's antistaphylococcal efficacy was conducted using the modified broth microdilution method specifically designed for volatile antimicrobial agents [32]. Additionally, its chemical composition was characterized by GC-MS and GC-QTOF-MS. The results of the broth microdilution volatilization method (Table 1) showed that *S. aureus* was susceptible to the *C. mangga* EO, exhibiting in vitro growth-inhibitory effect in the liquid phase at an MIC of 256 μg/mL. Still, no inhibition activity was observed during the vapour phase (MIC > 1024 μg/mL).

**Table 1.** In vitro antimicrobial activity of *Curcuma mangga* rhizome essential oil in liquid and vapour phase tested in broth and agar media, respectively.

| Microorganisms | Minimum Inhibitory Concentration (µg/mL) | | | |
| --- | --- | --- | --- | --- |
| | *Curcuma mangga* | | Tetracycline | |
| | **Broth** | **Agar** | **Broth** | **Agar** |
| *Staphylococcus aureus* ATCC 29213 | 256 | >1024 | 0.5 | ND |
| *Enterococcus faecalis* ATCC 29212 | >1024 | >1024 | 16 | ND |
| *Pseudomonas aeruginosa* ATCC 27853 | >1024 | >1024 | 16 | ND |
| *Escherichia coli* ATCC 25922 | >1024 | >1024 | 1 | ND |
| *Candida albicans* ATCC 10231 | >1024 | >1024 | 0.5 * | ND |

ATCC, American type culture collection; MRSA, methicillin-resistant *S. aureus* clinical isolate; TRSA, tetracycline-resistant *S. aureus*; *, Tioconazole for *C. albicans*; ND, Not determined.

The EO did not exhibit inhibitory activity against other microorganisms (*C. albicans*, *E. faecalis*, *E. coli*, and *P. aeruginosa*) in both liquid and vapour phases (MIC > 1024 µg/mL, respectively). In the modified broth microdilution method, *C. mangga* EO exhibited anti-staphylococcal activity at the MIC ranging from 128 to 1024 µg/mL (Table 2).

**Table 2.** In vitro antistaphylococcal activity of *Curcuma mangga* rhizome essential oil determined by EVA capmat cover modified method.

| Strains | Minimum Inhibitory Concentration (µg/mL) | |
| --- | --- | --- |
| | *Curcuma mangga* | Tetracycline |
| ATCC 29213 | 256 | 0.5 |
| ATCC 33591 | 1024 | 128 |
| ATCC 33592 | 512 | 128 |
| ATCC BAA-976 | 512 | 0.25 |
| MRSA 1 | 256 | 0.5 |
| MRSA 2 | 256 | 64 |
| MRSA 3 | 256 | 32 |
| MRSA 4 | 256 | 0.5 |
| TRSA 1 | 256 | 16 |
| TRSA 2 | 128 | 16 |

ATCC, American type culture collection; MRSA, methicillin-resistant *S. aureus* clinical isolate; TRSA, tetracycline-resistant *S. aureus*.

Of particular note is the heightened sensitivity of the tetracycline-resistant *S. aureus* (TRSA) 2 strain to the EO, displaying the lowest MIC of 128 µg/mL. This observation aligns with findings reported by Kamazeri et al. [28], where the *C. mangga* rhizome EO demonstrated growth-inhibitory activity against *S. aureus* ATCC 25923 at an MIC of 1.2 µL/mL. Furthermore, these results find support in studies conducted by Romulo [35] and Renisheya [36], both indicating the inhibitory activity of *C. mangga* rhizome extracts against *S. aureus.* However, what sets this study apart is its novel contribution—the demonstration of the EO's antistaphylococcal effect against a broad spectrum of standard and clinical strains, encompassing both methicillin-resistant *S. aureus* (MRSA) and tetracycline-resistant S. aureus (TRSA). This signifies a valuable extension of our understanding of the potential therapeutic applications of *C. mangga* EO in addressing infections caused by diverse strains, particularly those resistant to conventional antibiotics.

The hydrodistillation of *C. mangga* rhizome from Indonesia resulted in yellow oil yielding 0.32% (*v/w*). Fifty-two compounds were identified using HP-5MS and DB-17MS columns, representing 95.79/94.99% of their total contents (Table 3). The analyses showed that the major constituents of the oil from the *C. mangga* rhizome were represented by monoterpenoids, sesquiterpenoids, and diterpenoids, with a total content 89.57/79.45%. In contrast, the sesquiterpenoids were the most dominant components (43.82/34.38%).

**Table 3.** Chemical composition of *Curcuma mangga* rhizome essential oil.

| Component | Retention Indices (RI) | | Relative Content (%) | | Identification [c] |
|---|---|---|---|---|---|
| | Lit. [a] | Calc. [b] | HP-5 | DB-17 | |
| Fatty Acid | | | | | |
| Hexadecanoic acid | 1968 | 1975 | 3.37 | 12.03 | MS, RI |
| Group sum (%) | | | 3.37 | 12.03 | |
| Carbonylic compounds | | | | | |
| 2-Nonanone | 1092 | 1094 | 0.03 | 0.25 | MS, QTOF, RI |
| 2-Undecanone | 1294 | 1296 | 0.08 | 0.50 | MS, RI |
| Group sum (%) | | | 0.11 | 0.75 | |
| Monoterpenoids | | | | | |
| α-Pinene | 939 | 939 | 0.38 | 0.85 | MS, QTOF, RI, Std |
| Camphene | 953 | 956 | 0.84 | 1.95 | MS, QTOF, RI |
| β-Pinene | 980 | 986 | 1.55 | 3.43 | MS, QTOF, RI |
| m-Cymene | 1026 | 1031 | 0.14 | 0.43 | MS, QTOF, RI, Std |
| Limonene | 1030 | 1029 | 0.10 | 0.28 | MS, QTOF, RI, Std |
| Perillene | 1101 | 1102 | 0.15 | 0.21 | MS, QTOF, RI |
| α-Campholenal | 1125 | 1127 | 0.14 | 0.22 | MS, QTOF, RI |
| Nopinone | 1137 | 1139 | 0.14 | 0.82 | MS, QTOF, RI |
| Pinocarveol | 1139 | 1140 | 0.12 | 1.11 | MS, QTOF, RI |
| Pinocarvone | 1164 | 1164 | 0.18 | - | MS, QTOF, RI |
| Borneol | 1167 | 1167 | 0.03 | 0.13 | MS, RI |
| Isogeranial | 1184 | 1178 | 0.10 | 0.13 | MS, RI |
| α-Terpineol | 1189 | 1195 | 0.13 | 0.07 | MS, QTOF, RI |
| Myrtenal | 1193 | 1198 | 0.81 | 0.92 | MS, QTOF, RI |
| Verbenone | 1205 | 1212 | 0.10 | 0.14 | MS, QTOF, RI |
| Carveol | 1217 | 1220 | 0.18 | - | MS, QTOF, RI |
| Myrtenyl acetate | 1235 | 1243 | 0.12 | 0.17 | MS, QTOF, RI |
| Pinocarvyl acetate | 1258 | 1255 | 0.12 | - | MS, QTOF, RI |
| Bornyl acetate | 1285 | 1288 | 1.19 | 1.73 | MS, QTOF, RI |
| p-Mentha-1,4-dien-7-ol | 1296 | [d] | - | 0.41 | MS |
| Dihydro-β-ionone | 1433 | 1436 | 2.77 | - | MS, QTOF |
| Group sum (%) | | | 9.30 | 12.98 | |
| Sesquiterpenoids | | | | - | |
| Cryprotene | 1345 | 1355 | 0.51 | - | MS, QTOF, RI |
| Cyclosativene | 1368 | 1367 | 0.47 | 0.97 | MS, QTOF, RI |
| Sativene | 1396 | 1393 | 0.29 | 0.30 | MS, QTOF, RI |
| Selinane | 1464 | 1465 | 0.23 | - | MS, RI |
| Caparratriene | 1493 | 1494 | 1.34 | - | MS, RI |
| Cubebol | 1515 | 1520 | 0.11 | 0.03 | MS, QTOF, RI |
| Caryophyllene oxide | 1581 | 1588 | 3.90 | 6.06 | MS, QTOF, RI |
| Humulene oxide | 1606 | 1614 | 0.11 | - | MS, RI |
| 13-nor-Eremophil-1(10)-en-11-one | 1629 | 1635 | 7.19 | 6.21 | MS, QTOF, RI |
| Longifolenaldehyde | 1668 | 1677 | 3.04 | 1.62 | MS, QTOF, RI |
| Allohimachalol | 1674 | [d] | - | 1.88 | MS |
| Cadalene | 1674 | 1684 | 0.28 | - | MS, QTOF, RI |
| Alloaromadendrene oxide | 1646 | 1643 | 5.98 | 3.77 | MS, RI |
| Isolongifolol | 1738 | 1734 | 3.84 | 2.20 | MS, RI |
| Ambrial | 1809 | 1807 | 16.12 | 10.97 | MS, QTOF, RI |
| Dehydrosaussurea lactone | 1838 | 1838 | 0.29 | 0.37 | MS, QTOF, RI |
| Corymbolone | 1899 | 1899 | 0.12 | - | MS, RI |
| Group sum (%) | | | 43.82 | 34.38 | |
| Diterpeneoids | | | | | |
| 8,13-Epoxy-15,16-dinorlab-12-ene | 1873 | 1886 | 0.36 | 1.52 | MS, RI |
| Cembrene A | 1942 | 1939 | 0.47 | 0.67 | MS, QTOF, RI |
| m-Camphorene | 1960 | 1962 | 3.41 | - | MS, QTOF, RI |
| 15,16-Dinorlabda-8(17),12-dien-14-al | 1967 | 1964 | 6.61 | 11.57 | MS, QTOF, RI |
| 15,16-Dinorlabda-8(17),11-dien-13-one | 1994 | 1995 | 24.63 | 15.78 | MS, QTOF, RI |

**Table 3.** *Cont.*

| Component | Retention Indices (RI) | | Relative Content (%) | | Identification [c] |
|---|---|---|---|---|---|
| | Lit. [a] | Calc. [b] | HP-5 | DB-17 | |
| Manool | 2056 | 2047 | 0.51 | 0.39 | MS, RI |
| Labda-8(17),12-diene-15,16-dial | 2382 | 2376 | 0.48 | 2.20 | MS, QTOF, RI |
| Group sum (%) | | | 36.46 | 32.13 | |
| Others | | | | | |
| 1,3,3-Trimethyl-2-(2-methylcyclopropyl)-1-cyclohexene | [d] | 1281 | 0.49 | 1.03 | MS |
| 5,5,8a-Trimethyldecalin-1-one | [d] | 1541 | 0.28 | 0.32 | MS |
| 2-Methyl-4-(2,6,6-trimethyl-1-cyclohexen-1-yl) butanal | [d] | 1561 | 1.85 | 1.05 | MS, QTOF |
| 2-Methyl-4-(2,6,6-trimethyl-1-cyclohexen-1-yl)-2-butenal | 1584 | 1592 | 0.13 | 0.32 | MS, QTOF, RI |
| Group sum (%) | | | 2.75 | 2.72 | |
| Total Identified | | | 95.79 | 94.99 | |

(a) Retention indices taken from the literature [33,34]; (b) RI, retention indices calculated on DB-5 capillary column. (c) MS, identification based on mass spectra matching (MSD quadrupole); QTOF, identification based on mass spectra matching (QTOF-MS); RI, identification based on retention index; Std, identification based on co-injection of authentic standard; (d) data not available.

In *C. mangga* rhizome EO, 15,16-dinorlabda-8(17),11-dien-13-one (24.63/15.78%) was the main compound, followed by ambrial (16.12/10.97%), 13-nor-eremophil-1(10)-en-11-one (7.16/6.21%), 15,16-dinorlabda-8(17),12-dien-14-al (6.61/11.57%), and aromadendrene oxide (5.98/3.77%). There were also significant amounts of caryophyllene oxide (3.90/6.06%), camphene (0.84/1.95%), isolongifolol (3.84/2.20%), m-camphorene (3.41/-%) and hexadecanoic acid (3.37/12.03%).

The composition of *C. mangga* rhizome essential oil (EO) has exhibited notable variations in this study compared to previous reports, highlighting the influence of factors such as geographical origin, growing conditions, and isolation methods. β-pinene, identified as the predominant compound in the EO extracted from wild populations in Indonesia (>15%) [37], was found to constitute a minor percentage in the current study (<3%). Conversely, Kamazeri [28] reported caryophyllene oxide, a sesquiterpenoid, as the principal component in *C. mangga* rhizome EO obtained through steam distillation, constituting 18.71%. However, our study detected this compound in lower amounts, accounting for less than 6% of the EO composition. The uncommon dinorlabdanic diterpenoids have not yet been described in *C. mangga* essential oil. 15,16-dinorlabda-8(17),11-dien-13-one and other structurally similar diterpenes were isolated from the methanol and acetone extracts of the rhizomes of *Alpinia calcarata* [38] and *Alpinia formosana* [39], genus Alpina, belonging to the Zingiberaceae family. 13-nor-eremophil-1(10)-en-11-one was previously found in wood oil from *Xanthocyparis vietnamensis* known as Vietnamese gold cypress [40], and as a component of oil of fragrant grass vetiver (*Vetiveria zizanioides*, Gramineae) together with many other sesquterpens [41].

This shared presence of certain diterpenoids across different plant species within and outside the Zingiberaceae family underscores the potential conservation of these compounds with different biological significance.

Ambrial, another compound of interest, emerged as one of the major constituents in the EO of *C. mangga* rhizome from Indonesia. Intriguingly, only a minimal quantity (2.3%) of this compound was reported in the literature about plant material obtained in Malaysia [25]. These observed disparities underscore the complex nature of the chemical compositions in *C. mangga* EOs and emphasize the need to consider multiple variables in their analysis.

The pronounced differences between the chemical profiles presented in this study and the literature mentioned above could be attributed to several factors. Firstly, the geographical origin of the plants plays a pivotal role in determining the chemical composition of essential oils, as variations in soil composition, climate, and altitude can significantly impact the plant's secondary metabolite production. Additionally, growing conditions,

such as temperature, humidity, and sunlight exposure, can exert a substantial influence on the synthesis of specific compounds within the plant.

Furthermore, the isolation methods employed in EO extraction can contribute to variations in the final chemical profile. Different extraction techniques, such as hydrodistillation or steam distillation, may selectively capture certain compounds while excluding others, thereby influencing the overall composition of the essential oil.

In conclusion, the significant disparities observed in the chemical compositions of *C. mangga* EOs between this study and previous reports underscore the dynamic and multifaceted nature of plant chemistry. These variations offer valuable insights into the intricate interplay of environmental and methodological factors shaping the chemical profile of essential oils. Consequently, a comprehensive understanding of these factors is essential for elucidating the true therapeutic potential of *C. mangga* EO and ensuring its effective utilization in various applications. [42].

In order to enhance the precision in identifying the constituents of the essential oil (EO), two distinct columns, namely the non-polar HP-5MS and the middle-polarity DB-17MS, were employed. The result of using the DB-17MS column revealed 16 additional compounds identified in *C. mangga* rhizome EO. Enhancing the quantification and identification of essential oil (EO) components is crucial, and the utilization of dual-column/dual-detector systems contribute to improved accuracy and serves as a preventive measure against false-positive identifications of compounds. Employing such dual systems enhances the reliability of the analytical process, providing a more comprehensive and accurate depiction of the EO's composition. This approach mitigates the risk of misidentification, thereby fortifying the validity of results in EO analysis. [43]. Additionally, the sample was also analyzed using GC-QTOF-MS. This analytical approach provides fast scanning, higher sensitivity, and mass accuracy compared to the common quadrupole MS (GC-MS), as it can deconvolute the overlapping peaks and detect some minor compounds [44,45]. Based on this approach, 15 additional minor constituents were found in the *C. mangga* rhizome EO.

While the present study did not delve into the investigation of the antistaphylococcal activity of the main constituents of *C. mangga* rhizome EO and its underlying mechanisms, existing literature highlights the antimicrobial potential of some of the identified components. For instance, β-pinene has been previously recognized for its antimicrobial activity against *S. aureus* and methicillin-resistant *S. aureus* (MRSA) at minimum inhibitory concentrations (MICs) of 20 μL/mL and 6250 μg/mL, respectively [46,47]. Similarly, various studies have underscored the antimicrobial effects of essential oils containing sesquiterpenoids as their major components [48,49]. Caryophyllene oxide, a sesquiterpenoid detected in the *C. mangga* EO, has demonstrated antimicrobial activity against *S. aureus* at an MIC of 10.4 μg/mL in prior research [50]. It is plausible to speculate that these compounds identified in the EO may contribute to the observed antistaphylococcal activity of *C. mangga* rhizome EO in this study. To the best of our understanding, there is no existing research that documents the antimicrobial effects of the primary constituents of *C. mangga* rhizome essential oil in the vapour phase, namely 15,16-dinorlabda-8(17),11-dien-13-one, ambrial, 13-nor-eremophil-1(10)-en-11-one, 15,16-dinorlabda-8(17),12-dien-14-al, and aromadendrene oxide. Chromatograms of EOs analyzed on HP 5 and HP-17 columns can be seen in Figures 1 and 2. Spectra of the tree of the most abundant compounds 15,16-dinorlabda-8(17),11-dien-13-one, ambrial, 15,16-dinorlabda-8 (17),12-dien-14-al and 13-nor-Eremophil-1(10)-en-11-one are shown in Figures 3–6, respectively.

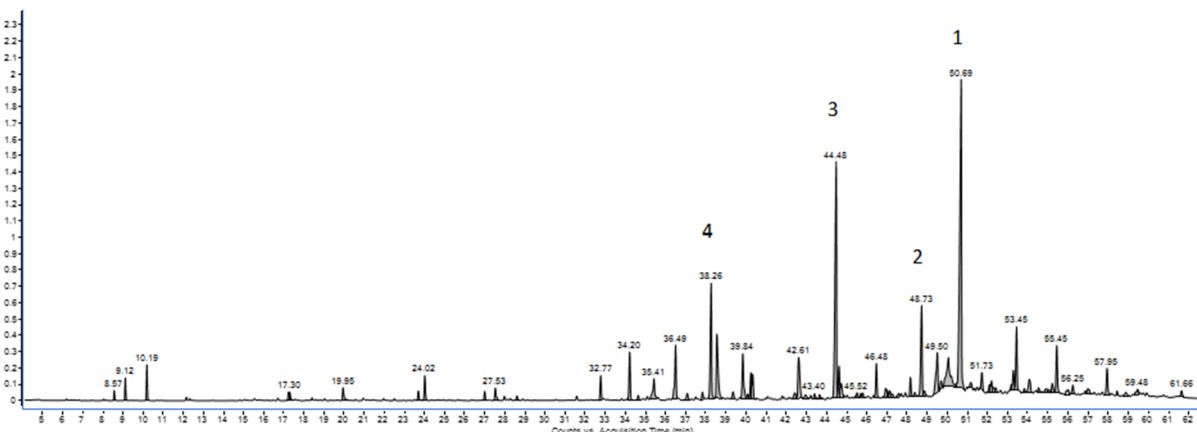

**Figure 1.** Chromatogram of *C. mangga* EOs analyzed on HP-5 column. Peak numbers and constituents' names: 1. 15,16-Dinorlabda-8(17),11-dien-13-one; 2. 15,16-Dinorlabda-8(17),12-dien-14-al; 3. Ambrial; 4. 13-nor-Eremophil-1(10)-en-11-one.

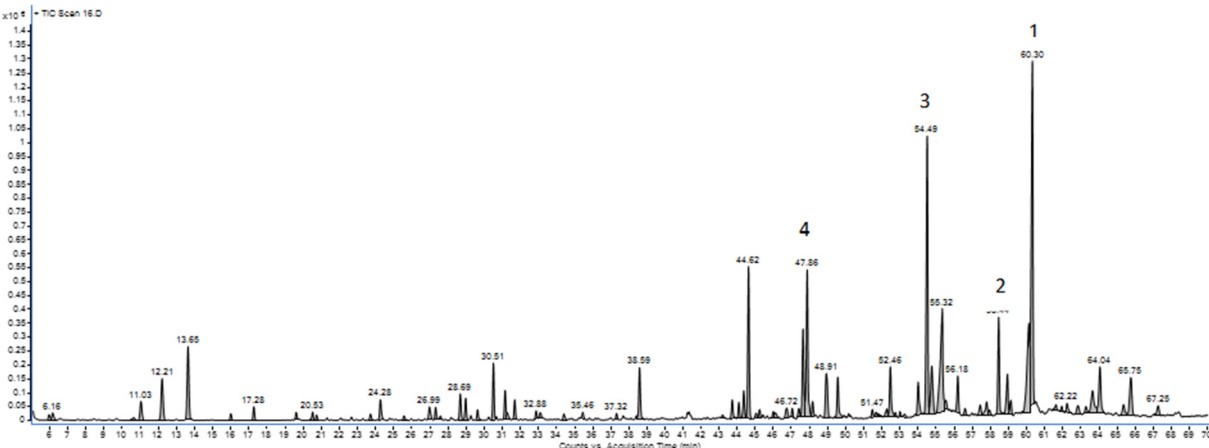

**Figure 2.** Chromatogram of *C. mangga* EOs analyzed on HP-17 column. Peak numbers and constituents' names: 1. 15,16-Dinorlabda-8(17),11-dien-13-one; 2. 15,16-Dinorlabda-8(17),12-dien-14-al; 3. Ambrial; 4. 13-nor-Eremophil-1(10)-en-11-one.

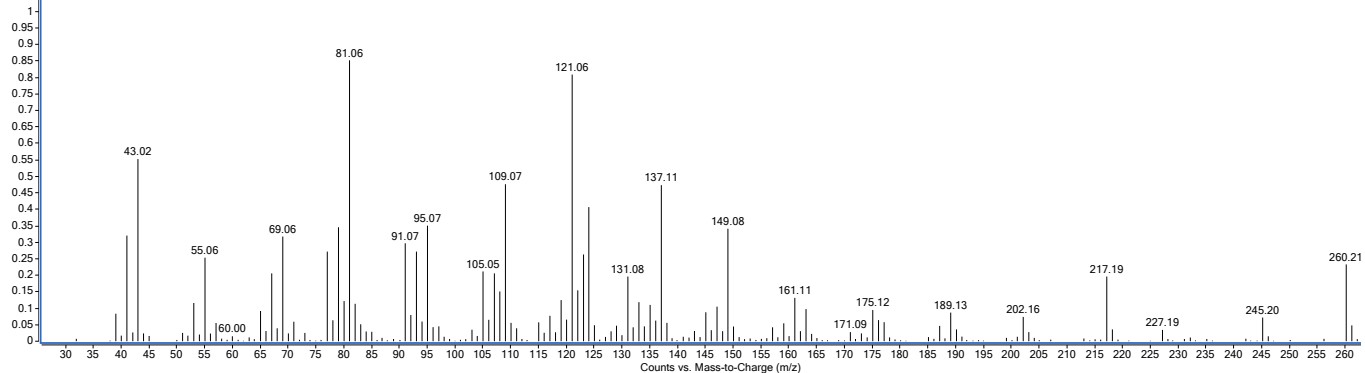

**Figure 3.** Mass spectrum of 15,16-Dinorlabda-8(17),11-dien-13-one (CAS Number, 76497-69-3).

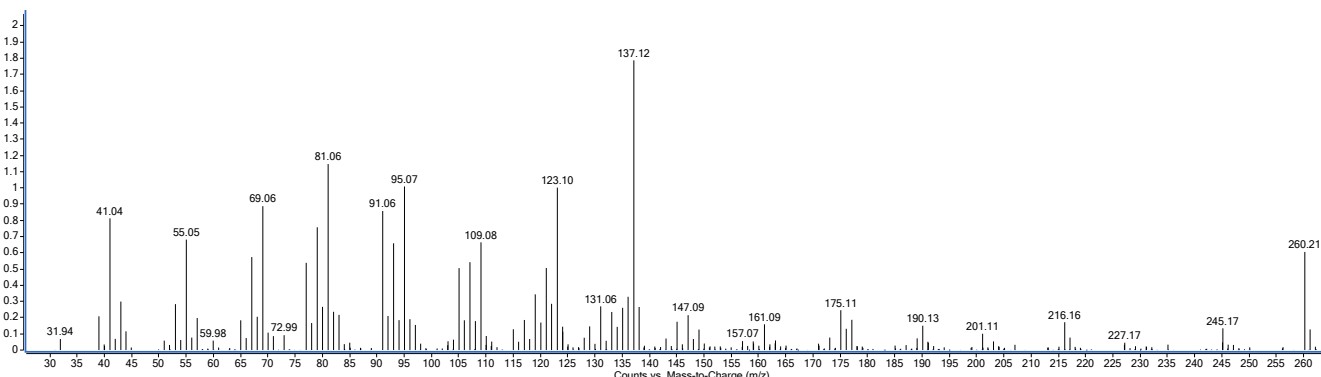

**Figure 4.** Mass spectrum of 15,16-Dinorlabda-8(17),12-dien-14-al (CAS Number, 167817-63-2).

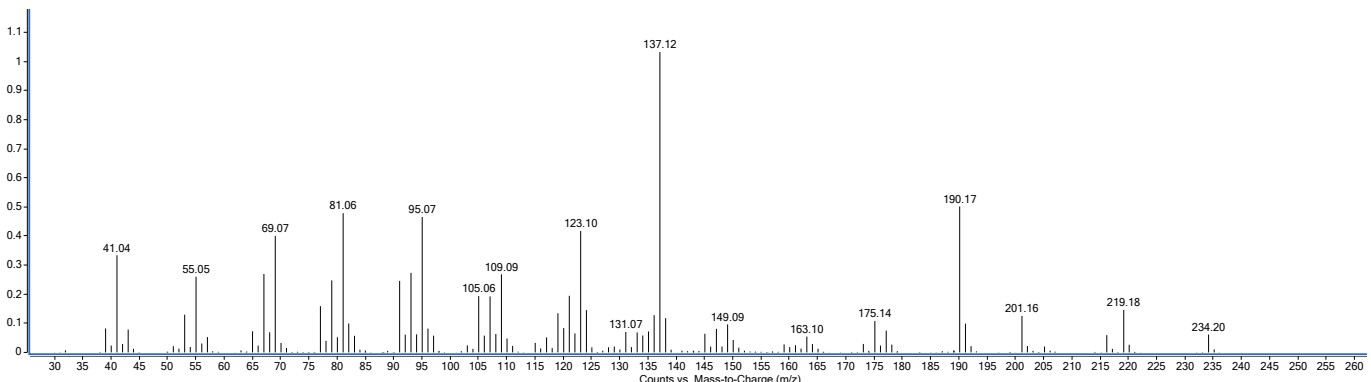

**Figure 5.** Mass spectrum of ambrial (CAS Number, 3243-36-5).

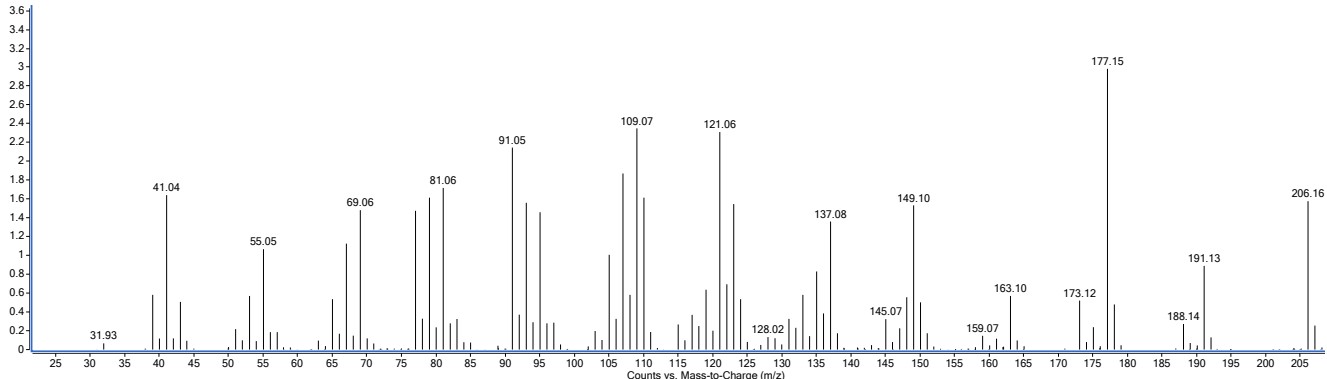

**Figure 6.** Mass spectrum of 13-nor-Eremophil-1(10)-en-11-one (CAS Number, 54275-21-7).

## 4. Conclusions

In this comprehensive article, we present a thorough analysis of the chemical composition and antistaphylococcal activity exhibited by the essential oil (EO) extracted from the rhizome of *C. mangga*, cultivated in Indonesia. Our findings underscore the significant antistaphylococcal potential of *C. mangga* EO, demonstrating efficacy against a diverse spectrum of *S. aureus* strains, encompassing antibiotic-resistant variants such as MRSA and TRSA.

Utilizing advanced analytical techniques, including gas chromatography–mass spectrometry (GC-MS) and gas chromatography–quadrupole time-of-flight mass spectrometry (GC-QTOF-MS), we meticulously identified key compounds present in the EO. Among these, 15,16-dinorlabda-8(17),11-dien-13-one, ambrial, 13-nor-eremophil-1(10)-en-11-one, 15,16-dinorlabda-8(17),12-dien-14-al, and aromadendrene oxide emerged as the major constituents.

The robust antistaphylococcal activity observed in our study positions *C. mangga* EO as a promising candidate for the development of anti-infective preparations rooted in traditional medicine. However, a cautious approach is warranted, acknowledging the preliminary nature of our findings. The journey from the current promising insights to a dependable therapeutic agent is multifaceted and requires further exploration.

Our study serves as a catalyst for future research endeavours, emphasizing the need for additional investigations into the safety profile and in vivo efficacy of *C. mangga* EO. Before considering its potential application in clinical settings, it is imperative to conduct rigorous assessments to ensure safety and efficacy.

In conclusion, while our study unveils promising avenues for leveraging *C. mangga* rhizome EO in combatting staphylococcal infections, the translation of these findings into a reliable therapeutic agent necessitates a holistic approach. Ongoing research initiatives, encompassing safety evaluations and in-depth efficacy studies, will pave the way for the integration of *C. mangga* EO into traditional medicine-based anti-infective preparations. By doing so, we contribute to the global efforts to address the challenges posed by antibiotic-resistant pathogens.

**Author Contributions:** Conceptualization, K.U. and A.R.; methodology, K.U., A.R., M.H. and P.N.; validation, L.K.; investigation, M.H. and A.R.; resources, K.U. and P.N.; writing—original draft preparation, A.R. and K.U.; writing—review and editing, L.K. All authors have read and agreed to the published version of the manuscript.

**Funding:** Czech University of Life Sciences Prague Internal Grant Agency project IGA 20233101.

**Data Availability Statement:** The data presented in this study are available within the article.

**Acknowledgments:** We thank Taopik Ridwan for his assistance in collecting the plant material.

**Conflicts of Interest:** The authors declare no conflicts of interest.

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
