# Peer review of "Chemical Composition and Antistaphylococcal Activity of Essential Oil of Curcuma mangga Rhizome from Indonesia"

_separations, doi:10.3390/separations11020049_

Round 1

Reviewer 1 Report

Comments and Suggestions for Authors

The article written by the authors presents an investigation into the staphylococcal properties of essential oil (EO) derived from the rhizome of Curcuma mangga, cultivated in Indonesia. Employing both the broth microdilution volatilization method and a modified standard broth microdilution method, the researchers evaluated the EO's effectiveness against Staphylococcus aureus, employing GC-MS to identify its chemical composition.

The authors employed multiple laboratory methods to do an investigation, studying both the essential oil composition and its antimicrobial properties. The authors included required control samples. The research is methodologically sound. The findings suggest promising applications for studied EO in combating staphylococcal infections. However, further research focused on its safety and in vivo efficacy will be needed before its possible use.

The article is written in clear and precise language.

There are some minor errors found and below are suggestions for authors to include, without diminishing the value of their work. These are:

Line 99 – ml>>> mL

Line 137 – It is a convention to italicize the letter "n” in “n-hexane”, it also relates to the analogical situation in the rest of the article (and para-, meta-, etc.)

In lines 182-184, the authors indicate the main identified components. These components have not been previously identified by other authors, and there is observable variation in the published compositions of essential oil (EO). In my view, this aspect warrants more extensive discussion. Also, the discussion about how certain/reliable are GC-MS identification done by authors:  15,16-dinorlabda-8(17),11-dien-13-one (24.63/15.78 %), ambrial (16.12/10.97%), 13-nor-eremophil-1(10)-en-11-183 one (7.16/6.21%), 15,16-dinorlabda-8(17),12-dien-14-al (6.61/11.57 %). It would be great if the authors revealed also CAsnames (for such unusual identifications) and the obtained MS library match % and the info if retention index reference values they took from reference [28] are real experimental data or estimations made by software. Maybe they can present some other literature data for these analytes? Given the inherent challenge in GC-MS identification of sesquiterpenes, transparency in presenting raw, detailed data, such as an exemplary chromatogram and mass spectrum of the main indicated components, would prove highly beneficial for future researchers exploring this topic (these could be included as supplementary files).

The authors transparently provided the details related to GC-MS analysis of the EO. They did also the analyses on the DB-17 column. It is a pity that the retention index (RI) values for DB-17 were omitted from Table 3 and not compared with reference values on this mid-polar GC column selectivity, especially for analytes where such a comparison is feasible. This information could be useful for other researchers and also showed additional confirmation of the identifications made.

Authors wrote in lines 202-205: “Enhancing the quantification and identification of essential oil (EO) components is crucial, and the utilization of dual-column/dual-detector systems contributes to improved accuracy and serves as a preventive measure against false-positive identifications of compounds.”  I agree, but on the condition that identification is not done primarily on MS. Running the sample on 2 GC column selectivities and confirmation of the analyte retention time or retention index is considered as gold standard. Running the sample on 2 different column selectivities and identifying them based on MS, in my opinion, has no such relevance, as it is missing retention data. Therefore, I recommend the authors do retention index validation of identifications on the second column. It may be difficult as the reliable RI reference data on the (50%-phenyl)-methylpolysiloxane column are not so popular. In this goal, usually one non-polar column and the second polar – wax phase – are employed.

While not challenging the correctness of the identification, the article could significantly benefit from a more in-depth discussion of the GC-MS identifications, including transparency in raw data presentation and additional information on column selectivity for DB-17. This would enhance the robustness of the findings and contribute to the overall reliability of the study in the context of the diverse compositions reported for this essential oil.

Overall, this study presents a valuable contribution to the field of essential oil chemistry and their potential antibacterial usage.

Author Response

Dear Reviewer,

Thank you for your careful review of our manuscript. We appreciate your feedback and the time you spent on our manuscript. We have carefully reviewed all your comments and edited and added to the manuscript with the best of intentions. Furthermore, the manuscript was expanded to the required number of words, especially in the Introduction and Discussion sections. Major changes in the text are highlighted.

Reviewer 2 Report

Comments and Suggestions for Authors

Good luck 

Author Response

Dear Reviewer,

Thank you for your careful review of our manuscript. We appreciate your feedback and the time you spent on our manuscript. We have carefully reviewed all your comments and edited and added to the manuscript with the best of intentions. Furthermore, the manuscript was expanded to the required number of words, especially in the Introduction and Discussion sections. Major changes in the text are highlighted.

LC1: This is a standard microbiological method, specifically the adaptation with the plates covered by the EVA Capmat.

LC2: You are correct, it's fixed, line 168.

LC3: The review focuses on aspects of infection control in the post-2015 future.